# Mediterranean Dietary Patterns Related to Sleep Duration and Sleep-Related Problems among Adolescents: The EHDLA Study

**DOI:** 10.3390/nu15030665

**Published:** 2023-01-28

**Authors:** José Francisco López-Gil, Lee Smith, Desirée Victoria-Montesinos, Héctor Gutiérrez-Espinoza, Pedro J. Tárraga-López, Arthur Eumann Mesas

**Affiliations:** 1Health and Social Research Center, Universidad de Castilla-La Mancha, 16071 Cuenca, Spain; 2Department of Environmental Health, T.H. Chan School of Public Health, Harvard University, Boston, MA 02138, USA; 3Centre for Health, Performance and Wellbeing, Anglia Ruskin University, Cambridge CB1 1PT, UK; 4Faculty of Pharmacy and Nutrition, UCAM Universidad Católica San Antonio de Murcia, 30107 Murcia, Spain; 5Escuela de Fisioterapia, Universidad de las Américas, Quito 170504, Ecuador; 6Departamento de Ciencias Médicas, Facultad de Medicina, Universidad de Castilla-La Mancha, 02008 Albacete, Spain; 7Postgraduate Program in Public Health, Universidade Estadual de Londrina, Londrina 86057-970, Brazil

**Keywords:** eating healthy, sleep recommendations, sleep guidelines, Mediterranean diet, youths

## Abstract

Purpose: The aim of the present study was to examine the association of adherence to the Mediterranean Diet (MD) and its specific components with both sleep duration and sleep-related disorders in a sample of adolescents from the *Valle de Ricote* (Region of Murcia, Spain). Methods: This cross-sectional study included a sample of 847 Spanish adolescents (55.3% girls) aged 12–17 years. Adherence to the MD was assessed by the Mediterranean Diet Quality Index for Children and Teenagers. Sleep duration was reported by adolescents for weekdays and weekend days separately. The BEARS (Bedtime problems, Excessive daytime sleepiness, Awakenings during the night, Regularity and duration of sleep, and Sleep-disordered breathing) screening was used to evaluate issues related to sleep, which include difficulties at bedtime, excessive drowsiness during the day, waking up frequently during the night, irregularity, length of sleep, and breathing issues while sleeping. Results: Adolescents who presented a high adherence to the MD were more likely to meet the sleep recommendations (OR = 1.52, 95% CI 1.12–2.06, *p* = 0.008) and less likely to report at least one sleep-related problem (OR = 0.56, 95% CI 0.43–0.72, *p* < 0.001). These findings remained significant after adjusting for sex, age, socioeconomic status, waist circumference, energy intake, physical activity, and sedentary behavior, indicating a significant association of adherence to the MD with sleep outcomes (meeting sleep recommendations: OR = 1.40, 95% CI 1.00–1.96, *p* = 0.050; sleep-related problems: OR = 0.68, 95% CI 0.50–0.92, *p* = 0.012). Conclusions: Adolescents with high adherence to the MD were more likely to report optimal sleep duration and fewer sleep-related problems. This association was more clearly observed for specific MD components, such as fruits, pulses, fish, having breakfast, dairies, sweets, and baked goods/pastries.

## 1. Introduction

Two factors intrinsic to health are diet and sleep, which may well influence one another [1]. For instance, sleep exerts an integral role in emotional regulation, modulating affective neural systems and reprocessing recent emotional experiences [2], which could lead to unhealthy food intake [3]. Furthermore, whole diets rich in vegetables, fruits, pulses, and other sources of dietary melatonin and tryptophan seem to predict favorable sleep-related outcomes [1]. Getting enough sleep is essential for overall well-being and good health [4]. Adequate sleep is vital for proper brain function and physical health, including maintaining metabolism, controlling hunger, and ensuring the proper functioning of the immune, hormonal, and cardiovascular systems [5]. However, teens are particularly at risk for not getting enough sleep [6]. The overall percentage of sleep complaints and sleep problems in the young population is very high, reaching approximately 80% in certain parts of the world [7]. Ensuring proper sleep is crucial for promoting healthy growth and development in adolescents since a lack of sufficient sleep or inconsistent sleep patterns can lead to an increased risk of both physical and mental health issues [8,9].

Importantly, dietary behavior has been shown to influence sleep via the food groups consumed and their nutrient content, as well as the timing when they were ingested prior to going to bed [10]. Recently, there has been much scientific research focusing on the connection between diet and sleep with the aim of better understanding how certain aspects of diet can directly impact the quality of sleep [11]. Sleep is influenced not only by the calorie content of a diet but also by the specific types of macronutrients that are consumed, such as proteins, fats, and carbohydrates [12]. Therefore, consuming a high-carbohydrate diet and foods rich in tryptophan, melatonin, and plant-based compounds shows potential in enhancing both the length and quality of sleep [11]. Supporting this notion, using diet management to improve sleep has been identified as a feasible, affordable, and convenient strategy [13].

The Mediterranean diet (MD), a plant-based eating plan that is becoming increasingly popular around the world, is considered to be one of the healthiest types of diets [14]. Research suggests that individuals can gain many benefits by incorporating elements of this diet into their eating habits [15]. This eating pattern includes a great consumption of plant-based foods (vegetables, fruits, breads and other cereals, beans, seeds, nuts, and potatoes); seasonally fresh, minimally processed, and locally grown foods; olive oil intake as the main fat source; and a moderate intake of dairy products (mostly as yogurt and cheese), among other components [14]. In addition, MD has been proposed as a “gold standard” diet due to its numerous benefits for health and nutrition, richness in biodiversity, great sociocultural food value, low environmental impact, and positive economic impacts on local communities [16].

The available studies suggest that higher adherence to the MD is linked with adequate sleep duration and with some indicators of greater sleep quality. However, most of these studies have focused on adults [17], and this relationship has been less studied among adolescents [18,19]. A deeper comprehension of the specific dietary elements that have a direct association with sleep duration and sleep-related problems is of upmost importance to establish future intervention programs aimed at improving sleep outcomes among this population. Thus, the objective of the present study was to examine the association of adherence to the MD and its specific components with both sleep duration and sleep-related disorders in a sample of adolescents from the *Valle de Ricote* (Region of Murcia, Spain).

## 2. Materials and Methods

### 2.1. Study Design and Population

This study is cross-sectional and uses data obtained from the Eating Healthy and Daily Life Activities (EHDLA) study, which analyzed a sample of adolescents aged between 12 and 17 years from *Valle de Ricote* (Region of Murcia, Spain) as representative participants. This study involved three secondary schools and collected data during the 2021/2022 academic year. The detailed methods of the EHDLA study are available elsewhere [20]. Initially, 1378 adolescents (100.0%) were randomly chosen. Of them, 277 (20.1%) were excluded due to the lack of data on sleep-related outcomes. In addition, other participants were removed because of missing data on adherence to a Mediterranean diet (*n* = 157; 11.4%), waist circumference (*n* = 45; 4.1%), sedentary behavior and physical activity (*n* = 10; 1.1%), and energy intake (*n* = 42; 4.7%). Thus, a total of 847 adolescents (55.3% girls) were included in the analyses.

The following inclusion criteria were established: (1) aged between 12 and 17 years old; (2) registered and/or living in *Valle de Ricote*; and (3) provided consent by parents or legal guardians and assent by the student. Participants were excluded when they (1) were exempt from the Physical Education class at school, as data collection was conducted in the physical education lessons; (2) had any pathology that demanded special attention or contraindicated physical activity; or (3) were under pharmacological treatment due to a chronic medical condition.

To take part in this research, written consent was obtained from the parents or guardians of the selected adolescent participants, and they were provided with an information sheet outlining the aims of the study, along with the tests and questionnaires that would be given. The adolescents were also asked for their agreement to participate.

This research was granted ethical clearance by the Bioethics Committee of the University of Murcia (ID 2218/2018), the Ethics Committee of the Albacete University Hospital Complex, and the Albacete Integrated Care Management (ID 2021-85). In addition, it was conducted in compliance with the guidelines of the Helsinki Declaration.

### 2.2. Procedures

#### 2.2.1. Adherence to the Mediterranean Diet

The Mediterranean Diet Quality Index for Children and Teenagers (KIDMED) was used to evaluate adherence to the MD. This index has previously been validated in the young Spanish population [21]. The KIDMED includes a 16-question test and ranges from −4 to 12 points. Questions about unhealthy aspects of the MD are given a score of −1, and those about healthy aspects are given a score of +1. The total score is then divided into three categories: high MD (>8 points), moderate MD (4–7 points), and low MD (≤3 points).

#### 2.2.2. Sleep Recommendations and Sleep-Related Problems (Outcomes)

The duration of sleep was assessed by asking the adolescents to separately report their typical bedtime and wake-up time for both weekdays and weekend days as follows: “What time do you usually go to bed?” and “What time do you usually get up?”. The daily sleep duration was calculated by averaging the night-time sleep duration on weekdays and weekends, and the formula [(average nocturnal sleep duration on weekdays × 5) + (average nocturnal sleep duration on weekends × 2)]/7 was used. As per the guidelines of the US National Sleep Foundation, individuals who slept less than 9–11 h (for 12–13 years old) or 8–10 h (for 14–17 years old) were considered not to meet the recommended sleep guidelines [5]. In addition, the BEARS (Bedtime problems, Excessive daytime sleepiness, Awakenings during the night, Regularity and duration of sleep, and Sleep-disordered breathing) self-report questionnaire [22] was used to screen for sleep-related problems in the study since it has been shown to be an easy-to-use tool with good accuracy in detecting sleep-related problems [23].

#### 2.2.3. Covariates

Sex and age were self-reported by adolescents. Socioeconomic status was assessed with the Family Affluence Scale (FAS-III) [24]. The FAS-III score was calculated by the sum of the responses from six different items related to family (i.e., vehicles, bedrooms, computers, bathrooms, dishwater, and travels). The final score ranged from 0 to 13 points. A higher score indicates greater socioeconomic status. Based on standard protocols, the measurement of waist circumference was taken by using a measuring tape with constant tension to the nearest 0.1 cm at the navel level. The Youth Activity Profile Physical (YAP), a self-administered 7-day (previous week) recall questionnaire that contains 15 different items separated into three sections (i.e., activity outside of school, activity at school, and sedentary habits), was applied to determine physical activity and sedentary behavior among the adolescents [25].

### 2.3. Statistical Analysis

The descriptive data are presented in the form of numbers and percentages for categorical variables and mean and standard deviation for continuous variables. To examine the relationship between following the different components of the MD and the dietary pattern with sleep outcomes, the Chi-square test was used. Since preliminary analyses showed no interaction between sex and adherence to the MD in relation to meeting sleep recommendations or sleep-related problems (meeting sleep recommendations: *p* = 0.689; sleep-related problems: *p* = 0.638), we analyzed both sexes together. Binary logistic regression analyses were performed to determine the odds ratio (OR) and the 95% confidence interval (CI) of the relationship between adherence to the MD (i.e., low/moderate MD or high MD) and each sleep outcome (i.e., meeting sleep recommendations or reporting sleep-related problems). The Hosmer–Lemeshow test was used as a statistical goodness-of-fit test for the logistic regression models. Based on Peduzzi et al. [26], sample size calculations for logistic regression were performed. Thus, the following formula was applied: *N* = 10 *k*/*p*, where *p* indicates the smallest proportion of sleep-related outcomes (i.e., meeting the sleep recommendations (24.8%), no sleep-related problems (43.3%)) in our sample and *k* is the number of independent variables (*k* = 8; adherence to the MD, sex, age, socioeconomic status, waist circumference, energy intake, physical activity, and sedentary behavior). Following this recommendation, the sample sizes required for the analyses were 323 for sleep recommendations and 185 for sleep-related problems. Sex, age, socioeconomic status, waist circumference, energy intake, physical activity, and sedentary behavior were included as covariates [27,28]. All analyses were conducted using SPSS software (IBM Corp, Armonk, NY, USA) version 28.0 for Windows. Statistical significance was determined at a *p*-value of less than or equal to 0.05.

## 3. Results

The descriptive data of the study participants are reported in Table 1. The KIDMED mean score was 6.6 ± 2.5 points. The proportion of adolescents meeting the sleep recommendations was 24.8%. A total of 57.7% of the adolescents reported at least one sleep-related problem.

The proportion of the different KIDMED components met and MD adherence in relation to the meeting of the sleep recommendations or the presence of sleep-related problems are shown in Table 2. A higher proportion of daily fruit intake was reported by adolescents meeting the sleep recommendations (*p* < 0.016) and not reporting any sleep-related problem (*p* < 0.031). Similarly, having breakfast and not eating sweets were more frequent in those meeting sleep recommendations (*p* < 0.11) and not presenting any sleep-related problems (*p* < 0.14). Participants meeting the sleep recommendations showed a greater proportion of second daily fruit intake and pulse intake more than once a week than those who did not meet these recommendations. Furthermore, participants not reporting any sleep-related problems reported a higher proportion of fish intake, higher intake of dairy products for breakfast, and lower intake of commercially baked goods/pastries for breakfast in comparison with their counterparts reporting sleep-related problems (*p* < 0.05 for all). Last, adolescents meeting sleep recommendations and not reporting any sleep-related problem had a higher adherence to the MD (i.e., KIDMED score ≥ 8 points) (*p* < 0.05 in both cases).

Figure 1 depicts the unadjusted and covariate-adjusted probability of meeting sleep recommendations or reporting any sleep-related problem according to adherence to the MD. In unadjusted analyses, adolescents who presented a high adherence to the MD were more likely to meet the sleep recommendations (OR = 1.52, 95% CI 1.12–2.06, *p* = 0.008) and less likely to report at least one sleep-related problem (OR = 0.56, 95% CI 0.43–0.72, *p* < 0.001). These findings remained significant even after adjusting for sex, age, socioeconomic status, waist circumference, energy intake, physical activity, and sedentary behavior, indicating an independent association between adherence to the MD and sleep outcomes (meeting sleep recommendations: OR = 1.40, 95% CI 1.00–1.96, *p* = 0.050; at least one sleep-related problem: OR = 0.68, 95% CI 0.50–0.92, *p* = 0.012). The Hosmer–Lemeshow test indicated a good fit of the models (sleep recommendations: *p* = 0.817; sleep-related problems: *p* = 0.217).

## 4. Discussion

Overall, the findings from the present study suggest that greater adherence to the MD was related to higher odds of meeting sleep recommendations, as well as with lower odds of reporting sleep-related problems among adolescents. These results are consistent with a previous review including adults and adolescents by Scoditti et al. [17], which pointed out that higher adherence to the MD is linked with adequate sleep duration and with numerous markers of better sleep quality. The MD includes a balanced ratio of fat, carbohydrates, and proteins and a particularly high content of vitamins and polyphenols, mostly provided by the moderate-to-high amounts of fruits, nuts, vegetables, cereals, olive oil, and fish [15]. Thus, the potential interactions between these foods and nutrients may provide an explanation for the positive effects of the MD on sleep outcomes [17]. Conversely, consuming a lot of processed meat, saturated fat, sugary drinks, and foods that are not commonly found in the MD has been associated with poorer sleep quality and shorter sleep duration, as well as insomnia symptoms [18,29]. In addition, certain micronutrient deficiencies (i.e., low availability of tryptophan) have been related to suboptimal hormonal regulation provoking disrupted sleep [30]. Likewise, macronutrient imbalances have also been noted, although not consistently, to affect sleep (i.e., energy-rich foods such as fats or refined carbohydrates) [30]. For instance, one study among US adolescents showed that those not meeting the sleep guidelines consumed a smaller variety of foods in comparison with their counterparts who did [31]. It is possible that high adherence to MD avoids nutritional deficiencies through an adequate and balanced intake of micronutrients and macronutrients [14] and, therefore, a higher quality and duration of sleep can be achieved.

Importantly, our results depicted the crucial role of fruit intake on sleep outcomes. There are some possible reasons justifying this finding. Fruits are rich in vitamin C, which is an antioxidant that scavenges free radicals. A growing body of evidence supports that vitamin C may have a role in sleep health due to the association between free radical formation and oxidative stress with sleep and sleep-related problems [32]. In addition, fruits are rich in some B-complex vitamins, which are associated with sleep outcomes. An example of this is that vitamin B6 helps in the creation of serotonin, which is a neurotransmitter that influences mood and hunger, and it also plays a role in making melatonin, which is an important hormone that regulates sleep, through the conversion of 5-hydroxytryptophan [12]. Moreover, fruits usually contain a high amount of vitamin B9 (i.e., folic acid). Indeed, deficiency in this vitamin has been associated with insomnia and sleep disturbances, possibly because vitamin B9 seems to be involved in the conversion of tryptophan into serotonin [33]. Another alternative explanation may be the high fiber content of fruits, since low fiber intake has been related to lighter and less restorative sleep, with greater arousals [29].

Moreover, our results showed that skipping breakfast was associated with not meeting sleep recommendations and sleep-related problems. Consistent with this finding, skipping breakfast and inconsistent eating schedules have been closely related to poor sleep quality [12]. Similarly, another study among Greek adolescents depicted that those skipping breakfast had greater odds of not meeting the sleep recommendations [34]. Skipping breakfast seems to be related to higher psychosocial health problems among the young population [35], and increased exposure to these health problems has been linked with current and later sleep-related problems among adolescents [36], which could justify this finding.

In addition, in our results, adolescents who usually had commercially baked goods or pastries for breakfast or sweets and candy several times daily reported more sleep-related problems than those who had not. Similarly, the proportion of adolescents meeting the sleep recommendations was lower in those who ate sweets and candy frequently. One possible explanation is that these foods are ultra-processed foods (e.g., highly profitable, ready-to-consume, hyperpalatable) according to the NOVA classification [37], which have been linked to lower sleep quality and duration among adolescents [38]. As an example, a study conducted on Brazilian adolescents found that consuming a high amount of highly processed foods, such as sweets, was related to a greater likelihood of experiencing poorer sleep quality [39]. Furthermore, another study among Iranian female adolescents showed that ultra-processed food consumption was linked to greater odds of reporting insomnia [40]. Likewise, sugar and high saturated fat intake (characteristic of ultra-processed foods) are associated with less restorative sleep [29]. The intake of ultra-processed foods may be associated with increased levels of inflammatory biomarkers, which are related to harmful sleep outcomes [41]. Furthermore, the consumption of ultra-processed foods has been related to higher depressive symptoms [42], anxiety-induced sleep disturbance [43], and mental health complications [44], likely owing to various mechanisms (e.g., inflammation, neuroplasticity, hypothalamic–pituitary–adrenal axis function).

Another finding of this study was that adolescents who more frequently consumed fish, pulses, and dairy products (for breakfast) were less likely to have negative sleep outcomes than those who consumed these foods less frequently. This is possibly due to the healthy nutritional profile of these foods. For example, fish intake could provide more long-chain polyunsaturated fatty acids (e.g., omega-3), which has been related to better sleep health *per se* [45,46]. In addition, Jansen et al. [45] showed that plasma docosahexaenoic acid was associated with earlier sleep timing and longer weekend sleep duration in Mexican adolescents. In addition, a meta-analysis by Dai et al. [46] concluded that omega-3 (e.g., docosahexaenoic acid, eicosatetraenoic acid) could potentially enhance specific aspects of sleep well-being throughout the period of childhood. Concerning pulses, their high amount of tryptophan could (at least partially) explain our findings. For instance, beans contain a high amount of tryptophan, which is a type of amino acid, and the impact of beans on sleep is related to the role they play in the formation of the brain chemical called serotonin [8]. Tryptophan is utilized to produce serotonin, and this chemical is then converted into melatonin [47]. In addition, pulse intake may provide favorable changes in the ratio of tryptophan to other large neutral amino acids (LNAAs) (valine, isoleucine, leucine, tyrosine, phenylalanine, methionine, and histidine). This could lead to a greater transport of tryptophan to the brain [48] and, therefore, to better sleep health [30]. In relation to dairy products, our findings showed a lower proportion of at least one sleep-related problem in those adolescents consuming this type of food for breakfast. It is possible that adolescents who include dairy in their breakfasts could obtain higher levels of tryptophan (e.g., milk [49]), which likely improves some sleep-related parameters (e.g., decreased sleep latency and increased sleep time and efficiency) [50].

Our results must be interpreted while considering some limitations. Because this study used a cross-sectional design, it is not possible to draw a cause-and-effect relationship from the findings obtained. Further studies with different designs (e.g., experimental) are required to examine whether higher adherence to MD and its components reduces the risk of sleep disturbances in adolescents. Although clinical trials are needed to confirm a causal impact of dietary patterns on sleep and elucidate the underlying mechanisms, the available data illustrate a cyclical relation between these lifestyle factors [1]. Similarly, the use of questionnaires to gather information on MD and sleep outcomes may result in bias due to the potential for differences in the desire to provide information or inaccuracies in recalling information. The potential effects of melatonin obtained from exogenous sources (such as diet) on the sleep–wake cycle may vary depending on the time of day at which it is consumed (i.e., morning, afternoon, or evening) [51]. Therefore, because the timing of adolescents’ meals was not available, we cannot affirm that a better sleep profile in adolescents with greater adherence to the MD is due to the intake of foods rich in bioactive nutrients related to sleep, such as tryptophan and melatonin. Conversely, a strength of this study is that it uses a sample of adolescents from *Valle de Ricote* (Region of Murcia, Spain) which is representative and makes our results generalizable to a broader population. Another strength is that, to our knowledge, the results from this study offer cross-sectional evidence of the association between MD and sleep outcomes (i.e., sleep recommendations, sleep-related problems) in an understudied population (i.e., adolescents).

## 5. Conclusions

Adolescents with high adherence to the MD were more likely to report optimal sleep duration and fewer sleep-related problems. This association was more clearly observed for specific MD components, such as fruits, pulses, fish, having breakfast, dairy products, sweets, and baked goods/pastries. Prospective studies are required to evaluate whether promoting MD is an effective strategy to prevent negative sleep outcomes among this population.

## Figures and Tables

**Figure 1 nutrients-15-00665-f001:**
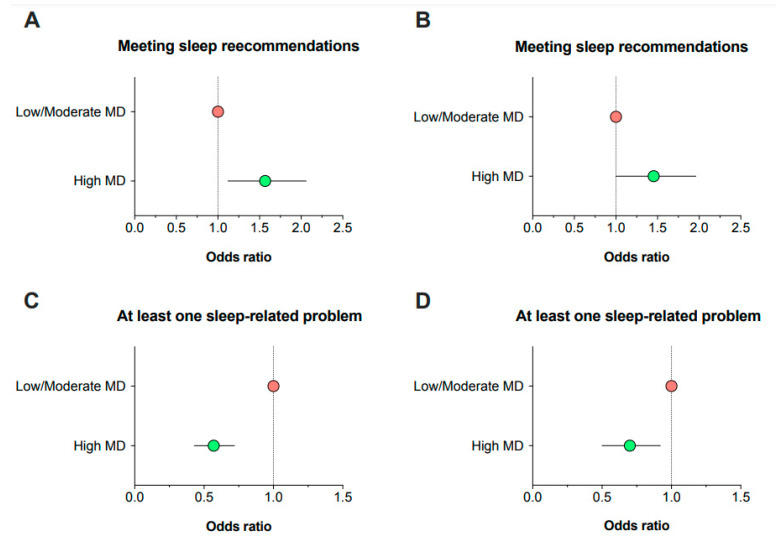
Odds ratio of meeting the sleep recommendations or reporting any sleep-related problem in relation to adherence to the Mediterranean diet. Data are expressed as dots (odds ratio) and lines (95% confidence intervals). (**A**): unadjusted; (**B**): adjusted for sex, age, socioeconomic status, waist circumference, energy intake, physical activity, and sedentary behavior; (**C**): unadjusted; (**D**): adjusted for sex, age, socioeconomic status, waist circumference, energy intake, physical activity, and sedentary behavior.

**Table 1 nutrients-15-00665-t001:** Characteristics of the study participants.

Variables	M ± SD/*n* (%)
Covariates	
Sex	
Boys	379 (44.7)
Girls	468 (55.3)
Age (years)	14.1 ± 1.6
FAS-III (score)	8.2 ± 2.1
Waist circumference (cm)	73.0 ± 10.3
Energy intake (kcal)	3027 ± 1980
YAP-S physical activity (score)	2.6 ± 0.7
YAP-S sedentary behavior (score)	2.6 ± 0.6
Adherence to the MD	
KIDMED (score)	6.6 ± 2.5
Low and moderate MD, n (%)	524 (61.9)
High MD, n (%)	323 (38.1)
Sleep duration	
Weekdays (min)	466.0 ± 60.1
Weekends (min)	555.3 ± 83.7
Overall (min)	491.5 ± 53.7
Meeting with sleep recommendations, n (%)	210 (24.8)
Sleep-related problems	
Bedtime problems, n (%)	195 (23.0)
Excessive day sleepiness, n (%)	281 (33.2)
Awakening during night, n (%)	128 (15.1)
Regularity and duration of sleep, n (%)	247 (29.2)
Sleep-disordered breathing, n (%)	56 (6.6)
At least one sleep-related problem, n (%)	489 (57.7)

Data are expressed as the mean (standard deviation) or count (percentages). BMI, body mass index; FAS-III, family affluence scale-III; KIDMED, Mediterranean Diet Quality Index for children and teenagers; MD, Mediterranean diet; YAP-S, Spanish Youth Active Profile.

**Table 2 nutrients-15-00665-t002:** Mediterranean Diet Quality Index for Children and Teenagers according to meeting with sleep recommendations or presenting at least one sleep-related problem.

Variables	Not Meeting Sleep Recommendations (*n* = 637; 75.2%)	Meeting Sleep Recommendations (*n* = 210; 24.8%)	*p*	No Sleep-Related Problems (*n* = 358; 43.3%)	At Least One Sleep-Related Problem (*n* = 489; 57.7%)	*p*
Individual components						
Takes a fruit or fruit juice every day	459 (72.1)	169 (80.5)	0.016	279 (77.9)	349 (71.4)	0.031
Has a second fruit every day	238 (37.4)	96 (45.7)	0.032	149 (41.6)	185 (37.8)	0.265
Has fresh or cooked vegetables regularly once a day	450 (70.6)	152 (72.4)	0.630	261 (72.9)	341 (69.7)	0.315
Has fresh or cooked vegetables more than once a day	210 (33.0)	75 (35.7)	0.465	109 (30.4)	176 (36.0)	0.092
Consumes fish regularly (at least 2–3 times per week)	371 (58.2)	136 (64.8)	0.095	229 (64.0)	278 (56.9)	0.037
Goes more than once a week to a fast-food (hamburger) restaurant	445 (69.9)	143 (68.1)	0.631	255 (71.2)	333 (68.1)	0.329
Likes pulses and eats them more than once a week	477 (74.9)	176 (83.8)	0.008	285 (79.6)	368 (75.3)	0.136
Consumes pasta or rice almost every day (5 or more times per week)	369 (57.9)	113 (53.8)	0.296	193 (53.9)	289 (59.1)	0.132
Has cereals or grains (bread, etc.) for breakfast	374 (58.7)	127 (60.5)	0.652	213 (59.5)	288 (58.9)	0.860
Consumes nuts regularly (at least 2–3 times per week)	373 (58.6)	122 (58.1)	0.907	217 (60.6)	278 (56.9)	0.272
Uses olive oil at home	622 (97.6)	335 (96.2)	0.325	352 (98.3)	472 (96.5)	0.136
Skips breakfast	108 (17.0)	19 (9.0)	0.005	39 (10.9)	88 (18.0)	0.004
Has a dairy product for breakfast (yogurt, milk, etc.)	483 (75.8)	164 (78.1)	0.502	287 (80.2)	360 (73.6)	0.027
Has commercially baked goods or pastries for breakfast	104 (16.3)	29 (13.8)	0.385	35 (9.8)	98 (20.0)	<0.001
Takes two yogurts and/or some cheese (40 g) daily	237 (37.2)	85 (40.5)	0.397	141 (39.4)	181 (37.0)	0.483
Takes sweets and candy several times every day	480 (75.4)	176 (83.8)	0.011	198 (55.3)	326 (66.7)	0.014
Adherence to the MD						
High MD (≥8 points)	226 (35.5)	97 (46.2)	0.006	160 (44.7)	163 (33.3)	<0.001

Data are expressed as numbers (percentages). MD, Mediterranean diet.

## Data Availability

Not applicable.

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
