# Peer review of "Mediterranean Dietary Patterns Related to Sleep Duration and Sleep-Related Problems among Adolescents: The EHDLA Study"

_nutrients, 2023, doi:10.3390/nu15030665_

Round 1

Reviewer 1 Report

Manuscript ID: nutrients-2184342

General remark

1. The authors use expressions suggesting a causal relationship between eating behavior and sleep function. Such formulations should be changed to neutral ones, since the design of the study does not allow them to judge the causal relationships between the studied parameters. Moreover, in the limitations section, the authors indicate that the design of their study is cross-sectional.

2. In the previous publication, the authors estimated the necessary and sufficient sample size of 1,138 people. The sample size in the submitted manuscript is only 75% of this value. The authors need to justify such a discrepancy between the obtained and the required sample size.

3. In addition, the authors conducted a regression analysis. Therefore, it is necessary to assess the compliance of the sample size for this type of analysis, taking into account the number of predictors in the models. If, in this case, the actual sample size does not meet the requirements for obtaining reliable results, the authors need to think about how to reduce the number of predictors in the models or conduct additional research to eliminate this shortcoming.

Introduction

4. In the introduction, the authors presented a review of the literature concerning only the effect of food components and diet on sleep function. However, there are numerous studies indicating reciprocal relationships between these indicators. It is necessary to provide references to the literature concerning the influence of sleep function on eating behavior and food preferences.

Methods

5. P.3, L.122-133. Replay.

6. It is necessary to provide information about checking the goodness of fit of the logistic regression models.

Results

7. The text of the manuscript mentions a figure, but the figure itself is missing.

Discussion

8. The authors discussed in detail the potential role of dietary tryptophan, which is a precursor of melatonin, in ensuring the maintenance of sleep function. However, it is known that melatonin affects sleep function differently when consumed before bedtime or in the morning after sleep. In the first case, it shifts the sleep phase to an earlier time of day, and in the second, on the contrary, it delays the sleep phase. In the study, the authors did not study the timing of adolescent meals, so it is necessary to be critical of such an explanation of the results or to provide references to publications on the half-life of tryptophan ingested with food, since there is no “depot” of tryptophan or melatonin in the body.

Reviewer 2 Report

see attached file

Round 2

Reviewer 1 Report

No comments.